# Current State and Future Trends: A Citation Network Analysis of the Academic Performance Field

**DOI:** 10.3390/ijerph17155352

**Published:** 2020-07-24

**Authors:** Clara Martinez-Perez, Cristina Alvarez-Peregrina, Cesar Villa-Collar, Miguel Ángel Sánchez-Tena

**Affiliations:** School of Biomedical and Health Science, Universidad Europea de Madrid, 28670 Madrid, Spain; cristina.alvarez@universidadeuropea.es (C.A.-P.); villacollarc@gmail.com (C.V.-C.); miguelangel.sanchez@universidadeuropea.es (M.Á.S.-T.)

**Keywords:** academic performance, citation network, motivation

## Abstract

*Background*: In recent years, due to its complexity and relevance, academic performance has become a controversial research topic within the health and educational field. The main purposes of this study were to analyze the links between publications and authors via citation networks, to identify the different research areas and to determine the most cited publications. *Methods*: The publication search was performed through the Web of Science database, using the term “Academic Performance” for a time interval from 1952 to 2019. The software used to analyze the publications was the Citation Network Explorer. *Results*: We found a total of 16,157 publications with 35,213 citations generated in the network, and 2018 had the highest number of publications of any year. The most cited publication was published in 2012 by Richardson et al. with a citation index score of 352. By using the clustering function, we found nine groups related to different areas of research in this field: health, psychology, psychosociology, demography, physical activity, sleep patterns, vision, economy, and delinquency. *Conclusions*: The citation network showed the main publications dealing with the different factors that affect academic performance, and it was determined that psychological and psychosocial factors were the most relevant.

## 1. Introduction

Academic performance is frequently associated with a country’s social and economic development [1]. Numerous studies have concurred that academic achievements are the result of cognitive and non-cognitive abilities, as well as the socio-cultural environment in which the learning takes place [2,3]. In this sense, academic performance is associated with intellectual factors, for example, long-term memory or the ability to think abstractly, and non-intellectual factors, such as motivation or self-discipline [4].

Motivation is an internal condition that influences behavior, i.e., mental power that helps people to meet their goals and achieve academic success [5,6,7]. To be able to motivate students, it is important to instill in them a greater desire to learn and make them enjoy studying. It has been proved that unmotivated students are not interested in learning or participating in class, and this attitude will affect their academic performance [6].

Scientific literature supports that cognitive abilities and self-efficacy can predict academic performance outcomes [8,9]. In 1983, Gardner explained [10] that self-sufficient people can organize themselves and carry out actions to meet their goals. 

Furthermore, psychosocial wellbeing is strongly related to thoughts, motivation, and decision-making relating to academic effort [11,12]. According to Vilar et al., a poor psychosocial status may be the result of a poor socio-economic situation, unfavorable family circumstances, or a bad relationship with classmates [13]. Students in these situations tend to have a negative attitude towards school and the learning process. This results in a low level of self-discipline and bad performance. In contrast, students with a positive psychosocial dynamic tend to have a high socioeconomic level and a good family relationship. They are also optimistic and have a good attitude at school. 

Other factors related to good performance at school include a healthy diet (rich in fiber and nutrients); practicing physical activity (as this increases metabolism, oxygenation and blood flow, delivering the hormones that promote neurological health); sleeping well (which improves cognitive functions, such as memory and learning) and good vision (visual impairment harms the development of motor skills, cognition, and language in child development) [14,15,16,17].

Citation network analysis is used to search for a specific topic within the scientific literature. By analyzing one publication, it is possible to find other relevant additional publications, to demonstrate, qualitatively, and quantitatively, the connections between articles and authors by creating groups [18]. Furthermore, it is also possible to quantify the most cited publications in each group, to study the development of a research field, or to focus the literature search on a specific topic [19,20].

Given the numerous factors that affect academic performance, this study aimed to identify the different areas of research and to discover the most cited publications. The connection between publications and research groups was also analysed using CitNetExplorer, software which enables the study of how the scientific literature in a particular field of has evolved.

## 2. Materials and Methods 

### 2.1. Data Source

The search was performed using the Web of Science (WOS) database, by entering the main descriptor: “Academic performance”, as this is the most commonly used expression in both fields, education and health. The search was limited to a topical search (TS) on article, keywords, and abstract, linked with the OR Table. Time was limited to the interval 1952–2019. 

Likewise, the Web of Science database allows the adding of references to its library when performing bibliographical searches directly in external databases or library catalogues.

With regards to the citation indices, the Social Sciences Citation Index, Science Citation Index Expanded and Emerging Sources Citation Index were used.

On the other hand, and due to the differences in citation styles among authors and institutions, CiteSpace software was used with a view to standardizing the data. The search and download date of the publications was the 25 April 2020.

### 2.2. Data Analyisis

All the publications were analyzed using the Citation Network Explorer software. This software is used for analyzing and visualizing the citation networks of scientific publications. It allows the researcher to download citation networks directly from Web of Science. Likewise, it makes it possible to manage citation networks which include millions of publications and related citations. In this way, a citation network comprising millions of publications can be initiated so a deeper analysis can then be performed in order to obtain a smaller subnetwork of 100 publications on the same topic. 

By using the citation score as an attribute, a quantitative analysis of the most cited publications within a specific time interval was performed. Through this, internal connections within the WOS database were quantified. By considering other databases, included in the WOS, the external connections were also quantified [20].

CitNetExplorer offers a number of techniques to analyze the citation networks of publications. The clustering function is achieved using the formula developed by Van Eck in 2012 [20].
(1)$$V\left(c_{1\;},\ldots ,c_{n}\right)=\underset{i<j}{\sum }\delta \;\left(c_{i},c_{j}\right)\left(s_{ij}-\gamma \right)$$

The clustering function was used to establish a group for each publication. This function grouped those publications with a greater level of association according to the citation networks [20].

Finally, the central publications were analyzed using the Identifying Core Publications function. The role of this function was to identify the publications that were considered as the core of the citation network and eliminate those which were considered insignificant. The number of connections was established by the researchers, meaning therefore that the greater this parameter was, the lower the number of central publications would be [20]. Thus, those publications that presented four or more citations within the citations network were considered.

On the other hand, the drilling down function was used, as it allows deep analysis to be carried out at different levels for each of the groups.

## 3. Results

The first articles on academic performance were published in 1952, so therefore it was decided that the selected time interval would be from 1952 to 2019. 16,157 publications and 35,213 citations inside the WOS were found. Out of all the publications, 77.62% were articles, 13.5% proceedings papers, 3.75% reviews, 2.16% meeting abstracts, 1.61% book chapters and the remaining 1.01% was comprised of editorial materials, letters, notes, book reviews, corrections, books, data papers, news items, reprints, amendments, additions and retracted publications.

The number of publications on this topic has increased significantly since 2015 (1952–2014: 46.77% of publications; 2015–2019: 53.23% of publications). 2018 was the year with the highest number of publications, with 1889 publications and 120 citations on the network (Figure 1).

Figure 2 shows the citation network and Table 1 all the details of the 20 most-cited publications. The first was an article by Richardson et al. [21], which was published in 2012 and boasted a citation index of 352. This work was a systematic review and meta-analysis of the years between 1997 and 2010 that analyzed 7167 articles on different demographic and psychosocial factors that influence academic performance in university students.

Out of the 20 most-cited articles, 11 dealt with the psychological factors that affect academic performance [21,22,23,24,25,26,27,31,32,36,37]. Two of the articles discussed how demographic and cognitive factors and personality can predict future academic performance [30,40]. Three articles addressed the benefits that healthy lifestyles have on academic performance [33,34,39]. The impact of sleep on academic performance was considered in three of the articles [29,35,38] and the final article focused on how the use of digital devices and social media affects academic performance [28].

### 3.1. Description of the Publications.

Academic performance is a multidisciplinary research field and the areas of education (36%) and psychology (28.24%) (Figure 3) are particularly worth mentioning. Figure 4 shows the 10 journals with the highest number of publications. 

The countries with the highest number of publications are the United States (39.10%), Spain (8.51%) and England (5.54%) (Figure 5).

### 3.2. Clustering Function

23 groups were found using the clustering function, with 9 of these groups containing a significant number of publications; however, the remaining 14 groups only reached 1.5% (Figure 6).

With regards to clustering parameters, a resolution value of 1.0 (default value in CitNetExplorer software) was considered, and the minimal publication size for each group was 500.

Table 2 shows the information from the citation networks on the nine main groups, ordered from largest to smallest according to their size.

In group 1, 3223 articles and 11,097 citations were found across the network. The most cited publication was the article by Richardson et al. [21], which was published in 2012 in the Psychological Bulletin, and it also ranked first among the 20 most cited publications. In this group, the different articles analyzed the impact of personality, cognitive ability, self-discipline, motivation, and demographic and psychosocial factors on academic performance (Figure 7).

In group 2, 1095 publications and 4407 citations were found across the network. The most cited publication in this group was the article by Hillman et al. [33], which was published in 2008 in Nature Reviews Neuroscience. In this article, the authors concluded that physical exercise leads to greater physical and mental health throughout life. The articles in group 2 dealt with the influence of visual impairments, such as uncorrected refractive errors, on academic performance. The articles in this group also analyzed the association between the number of hours spent watching television and obesity, as well as the positive impact that a healthy lifestyle, such as a healthy diet or practicing physical exercise, has on cognitive skills and academic performance (Figure 8).

In group 3, 971 publications and 2638 citations were found across the network. The most cited publication was the article by Ferguson et al. [30], which was published in 2002 in the British Medical Journal. This paper analyzed the requirements for predicting future academic performance, such as previous academic ability, personality, learning, or personal references. The common topic in this group was how personality, demographic data, and mental (depression, anxiety, and stress) and emotional (motivation) states influence academic performance. These papers also considered the academic selection process, focusing specifically on careers in the field of health (Figure 9).

In group 4, 776 publications and 1354 citations were found throughout the network. The most cited publication was the article by Fuligni et al. [41], which was published in 1997 in Child Development. This article analyzed how family background, parental attitudes, and peer support influence the academic performance of immigrant students. The common theme in this group was the influence of psychosocial and economic factors, such as family, teachers, peers, gender, or corrected refractive errors on academic performance (Figure 10).

In group 5, 734 publications and 1567 citations were found throughout the network. The most cited publication was the article by Crede et al. [42], which was published in 2010 in Review of Educational Research. In this publication, the authors concluded that class attendance has a positive impact on grades. The common subject in group 5 was the relationship between students who attend class and those who work, and the impact of this on academic performance (Figure 11).

In group 6, 665 publications and 1107 citations were found throughout the network. The most cited publication was an article by Duncan et al. [43], which was published in 2007 in Developmental Psychology. In this article, the authors analyzed cognitive, attention, and socio-emotional skills in terms of academic performance. The common topic in this group was the relationship between the said skills, drug use, bullying, and delinquency, and academic performance (Figure 12).

In group 7, 640 publications and 2407 citations were found throughout the network. The most cited publication was an article by Curcio et al. [29], which was published in 2006 in the journal Sleep Medicine Reviews. In this publication, the authors analyzed the impact of sleep on academic performance. The common topic addressed by this group was the impact of sleep and stress on academic performance (Figure 13).

In group 8, 619 publications and 1587 citations were found throughout the network. The most cited publication was an article by Dupaul et al. [44], which was published in 1991 in School Psychology Review. This study analyzed the academic performance of children with behavioral disorders. The common topic in this group was how hyperactivity disorders influence academic performance (Figure 14).

In group 9, 591 publications and 1772 citations were found throughout the network. The most cited publication was an article by Kirschner et al. [28], which was published in 2010 in Computers in Human Behavior. This study analyzed the negative impact of social networks, such as Facebook, on academic performance. The common subject in this group was the influence of digital devices and social networks on academic performance (Figure 15).

Table 3 shows a more detailed description of the oldest and most recent publications from the nine main groups.

Figure 16 shows that, after analyzing the relationship among the different groups by means of the drilling down function, no connection has been found between the main publications in different groups.

#### 3.2.1. Group 1—Subclusters

10 subclusters have been found (Figure 17), five of which boast a significant number of publications (Table 4). The remaining groups are relatively small with less than 200 publications and 1778 citation networks.

#### 3.2.2. Group 2—Subclusters

10 subclusters have been found (Figure 18), three of which boast a significant number of publications (Table 5). The remaining groups are relatively small with less than 90 publications and 464 citation networks.

### 3.3. Core Function

4660 publications with four or more citations (28.8% of all publications) and a citation network of 23,747 were found (Figure 19).

## 4. Discussion

The main databases, such as WOS and Scopus, allow for the exploration of citation networks. However, these databases do not offer a general overview of the connection between citations from a group of publications. Thus, their usefulness is limited to carrying out a systematic review of all existing literature. This is the reason why we used CitNetExplorer software to visualize, analyze, and explore citation networks of scientific publications. CitNetExplorer offers a more detailed analysis of the citation networks in comparison with other databases such as WOS or Scopus [20].

As our main objective was to analyze all the existing literature on the different factors that influence academic performance, we used the WOS database to perform our bibliographic search, given that it is the only database for which the search range begins in 1900. However, it is important to point out that WOS only accepts journals with an international presence, and these are only admitted following a rigorous selection process. 

In this way, the CitNetExplorer software allowed us to gather and analyze all of the literature available on academic performance up to the present date. Furthermore, the connection between the fields of study and the different research groups was also found by using citation network analysis.

In order to obtain the results, the following functionalities were used: the clustering functionality, as it allows the publications to be grouped by citation relationship; the drilling down functionality, which provides a deeper analysis of the existing bibliography for each of the groups; and the core publications functionality, which shows the most relevant publications, i.e., those which have a minimum citation number. Therefore, these functionalities enable the researcher to conduct a complete analysis of the studies published on a particular subject. 

In recent years, the number of publications on academic performance has significantly increased. In the past, research carried out on this topic focused predominantly on the influence of psychological factors on academic performance [80,81]. From the year 2000, the advent of digital devices brought about scientific research into their influence on academic performance. These devices were responsible for the growth of a more sedentary lifestyle which resulted in numerous studies on the impact of such behaviour. 

Since 2015, the number of publications on psychological and psychosocial factors has increased by 52.83% and they have taken a multidisciplinary turn. As such, multiple factors that affect academic performance, such as demographic, psychosocial, psychological, lifestyle, and mental and physical health factors, have been analysed. [82,83,84,85]. 

With regards to the authors, Kirk is worth mentioning in the early years, for having published over 70 papers on academic performance between 1952 and 1982. Currently, Rozelle and Salamonson are considered as two of the most prominent authors in this field of study, as they have published 40 and 33 papers respectively in the period between the years 2005 and 2010 and up to date.

2018 has been identified as a “key year”, given the large number of publications and the content of the documents, in which the main topic of study was the importance of mental and physical health on academic performance [86,87]. In the study conducted by Stajkovic et al. [86], published in 2018, the authors confirmed that awareness and emotional stability predict self-efficacy and this is positively related to academic performance. In the study by de Greeff et al. [87], published in 2018, it was determined that physical activity has a positive effect on attention, while long-term physical activity programs have a positive impact on executive functions, attention, and academic performance. Therefore, this study has confirmed that all of the factors that influence academic performance are interrelated, with psychological and psychosocial factors considered to be the most prevalent.

The countries with a greater number of publications are the United States, Spain and Britain. The educational systems of these countries are intense, which places students at a higher risk of anxiety and stress. Furthermore, and with regards to psychosocial factors, the students are more competitive. Consequently, many studies have focused on analyzing risk factors for academic performance in these countries. Nevertheless, it must be taken into consideration that, in the case of China, students are showing symptoms of anxiety and depression more often, and therefore the number of publications is expected to increase over the coming years. 

In the United States, the number of publications is significantly higher than in other countries. This is due to increased interest in the topics related to student learning. Likewise, the trend for the number of publications to increase in the field of education in countries like the United States or Britain has been associated with a combination of factors, such as being English-speaking countries and the possible connections among the various research groups in the scientific community [88,89].

With regards to the journals with the larger numbers of publications, it is worth mentioning that they are mainly Psychology and Educational Science journals, as psychological and psychosocial factors are those with the greatest influence. Moreover, the majority of the journals are from the United States and Britain.

Nowadays, students are more competitive and are therefore at a greater risk of suffering from stress. Several authors have defined stress as “an adverse reaction that people have to excessive pressures and demands placed on them. In other words, stress arises when people find themselves in an overwhelming situation and believe they are unable to cope with it” [90,91]. The study by Aafreen et al. [92], which was published in 2018, corroborated that students who are unable to manage stress are affected mentally, physically and emotionally and as such they tend to present anxiety and depression, which leads to a decline in academic performance. At the same time, the articles by Chapell et al. [36] and Cassady et al. [37], published in 2005 and 2002, respectively, focused on the relationship between stress and final grades. Both of the studies concluded that higher levels of anxiety were associated with lower grades and vice versa. That is to say, stress has a negative impact on academic performance and, likewise, the results demonstrated that female students present higher levels of anxiety than male students. Nowadays, the preparation of students is not only based on their studying for an undergraduate degree, but also on the idea that they at least have to have a master’s degree, which can lead to a higher prevalence of stress, anxiety and depression.

Health professionals recommend physical exercise and a balanced diet to prevent stress. Several studies have shown that people who exercise are less likely to suffer from anxiety and depression, and that stress symptoms can be improved by doing exercise [93,94]. Another study carried out in Spain among university students demonstrated that having a poor diet and being overweight are strongly associated with higher levels of stress, especially in women, and these factors have a negative impact on academic performance [95]. Consequently, scientific evidence reveals a significant relationship between academic performance and health status, therefore vision problems, nutrition, stress, obesity, anxiety, behavioral disorders and aggression are associated with a poor academic performance [96]. It should be noted that the personality of students with psychosocial well-being is characterized by determination and pragmatism in their academic efforts, therefore enabling them to establish high-level objectives that they pursue successfully [13].

In the future, health and welfare programs must be established in schools to reduce the incidence of behaviors that could have a detrimental effect on the physical and mental health of young people, as this could result in improved academic performance. In the study by Van Loon et al. [97], published in 2019, two school programs were created to improve the control of anxiety and social abilities. These programs were aimed at preventing mental health problems that have a negative impact on academic performance. It is also important to consider that as students get older they become more competitive, therefore meaning that they are more likely to suffer from physical and mental damage. As a result, the content of programs aimed at controlling stress in order to improve academic performance will be different in both primary and secondary education and at university.

This study provides a brand new approach which can guide researchers in the revision of the wealth of literature that exists on academic performance. Over the coming years, research in this field will continue to focus mainly on the impact of psychological factors on academic performance as a consequence of the constant increase in competitiveness among students. In this way, research questions based on this data posed by future studies will focus mainly on how psychological and psychosocial factors could be improved in order to increase academic performance.

Likewise, the number of studies on the impact of the use of devices on academic performance is also likely to increase, given that constant changes in lifestyle are becoming more and more frequent. With regards to citation network studies, they are also likely to become more numerous as it is the only method of analysis which provides a global overview of the different research fields within a particular subject. Moreover, the CitNetExplorer software facilitates the analysis of all existing studies on a given topic, thus allowing for detailed research to be conducted, which might change the way in which research is carried out in different fields of study.

## 5. Conclusions

In this study, 16,157 publications on academic performance were analysed, which were obtained from the Web of Science (WOS) database between 1952 and 2019. All of the publications were analysed using the Citation Network Explorer software, which makes it possible to visualize, analyze and explore the citation networks of scientific publications.

Academic performance may be influenced by various factors, including psychological, psychosocial, economic, environmental or personal factors. Self-sufficient students present learning and self-control skills which facilitate studying, and as a consequence increase their motivation. This means that they have the skills and willingness to learn. Likewise, the personality of the students with psychosocial wellbeing is often characterized by determination and pragmatism in academic effort, which enables them to set high-level goals for themselves, which they successfully pursue. On the other hand, scientific evidence suggests a significant relationship between academic performance and health, meaning that vision problems, nutrition, stress, obesity, anxiety, behavioral disorders or aggression are associated with poor academic performance. This is why it is necessary to establish health and wellbeing programs in schools, which are aimed at preventing behaviors that put the mental and physical health of young people at risk, and at improving their academic performance.

The citation network showed the main publications on the different factors that affect academic performance, and it was determined that psychological and psychosocial factors were the most relevant. This study offers a global overview of the number of publications for each of the years and field of study, as well as a general description of the 20 most-cited publications.

## Figures and Tables

**Figure 1 ijerph-17-05352-f001:**
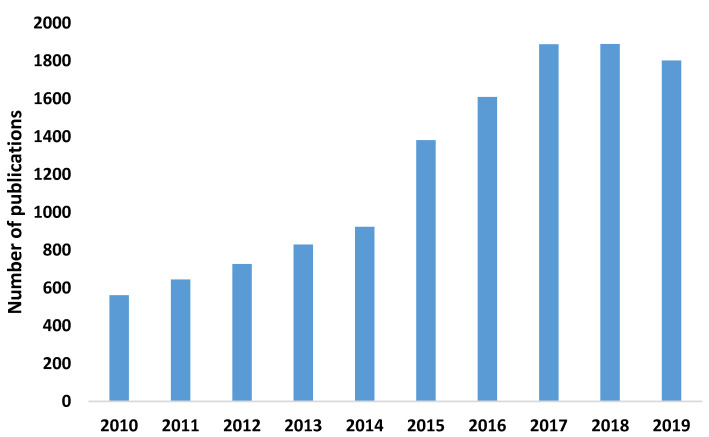
Number of publications per year.

**Figure 2 ijerph-17-05352-f002:**
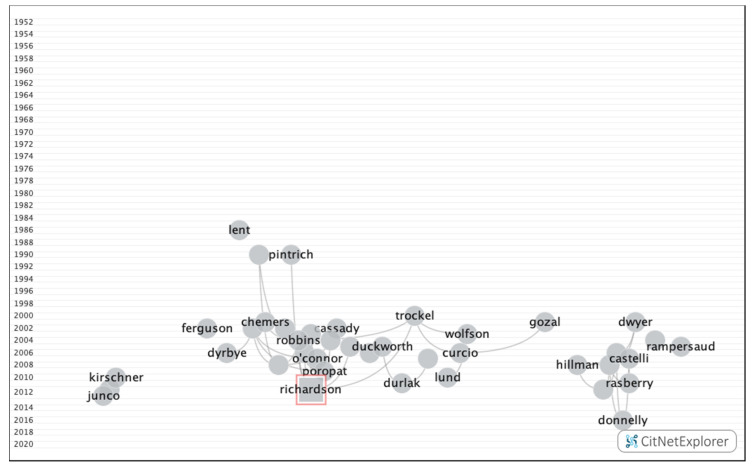
Citation Network of the 20 most cited publications on academic performance.

**Figure 3 ijerph-17-05352-f003:**
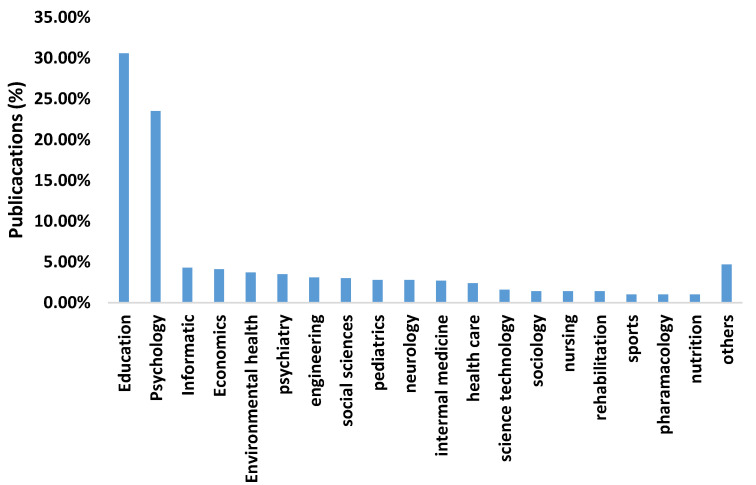
Percentage of publications by research field.

**Figure 4 ijerph-17-05352-f004:**
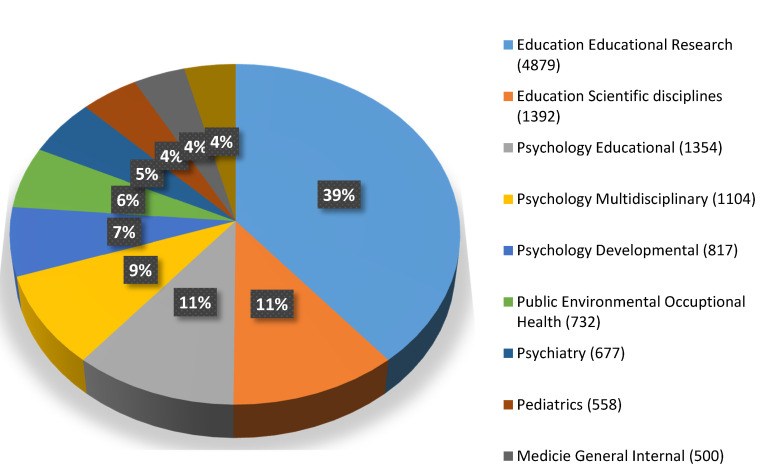
Top ten journals with the most publications.

**Figure 5 ijerph-17-05352-f005:**
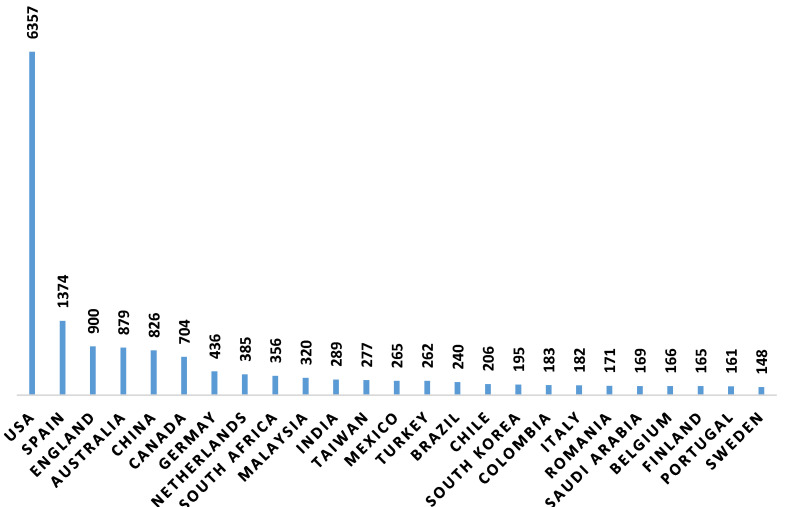
Number of publications by country.

**Figure 6 ijerph-17-05352-f006:**
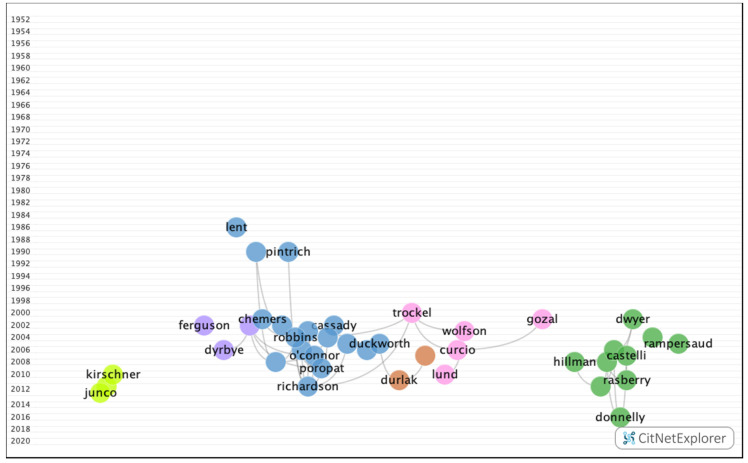
Clustering function in the network of citations on academic performance.

**Figure 7 ijerph-17-05352-f007:**
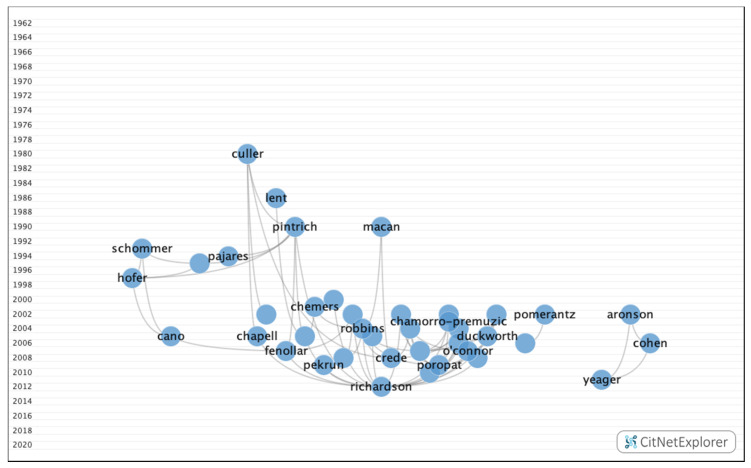
Group 1 citation network.

**Figure 8 ijerph-17-05352-f008:**
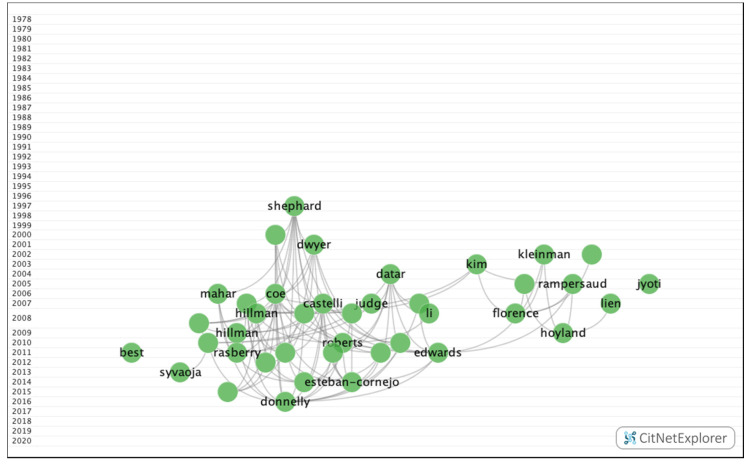
Group 2 citation network.

**Figure 9 ijerph-17-05352-f009:**
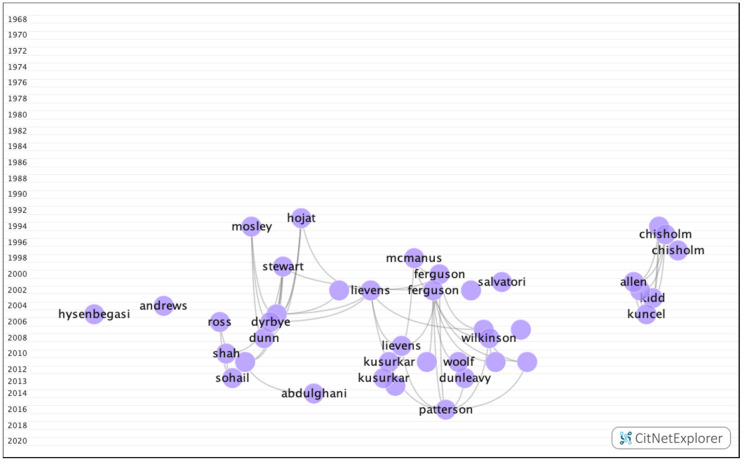
Group 3 citation network.

**Figure 10 ijerph-17-05352-f010:**
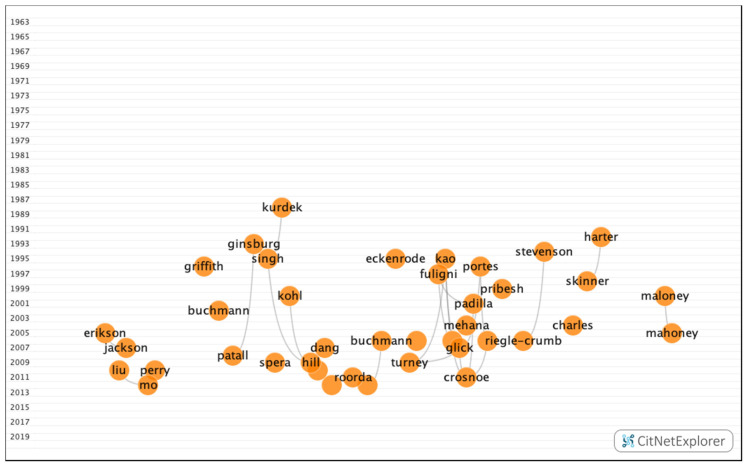
Group 4 citation network.

**Figure 11 ijerph-17-05352-f011:**
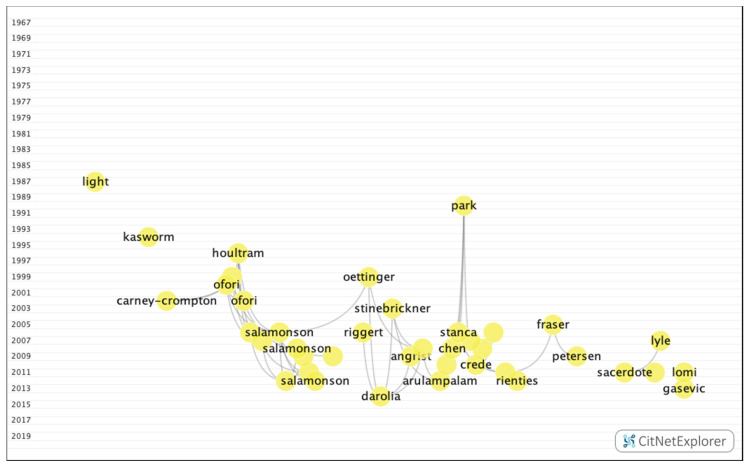
Group 5 citation network.

**Figure 12 ijerph-17-05352-f012:**
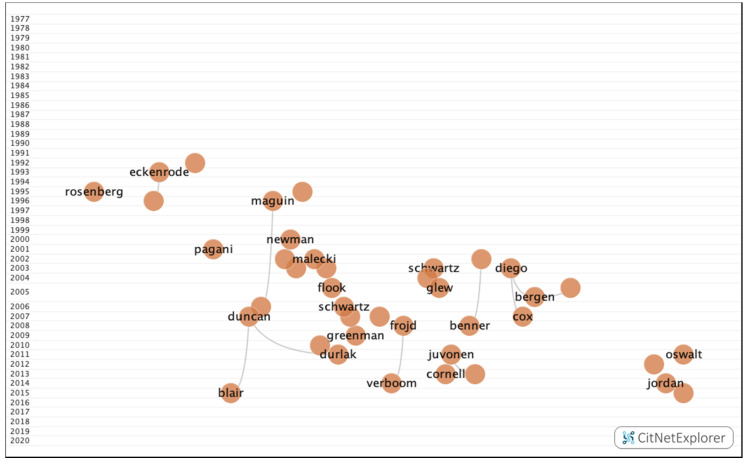
Group 6 citation network.

**Figure 13 ijerph-17-05352-f013:**
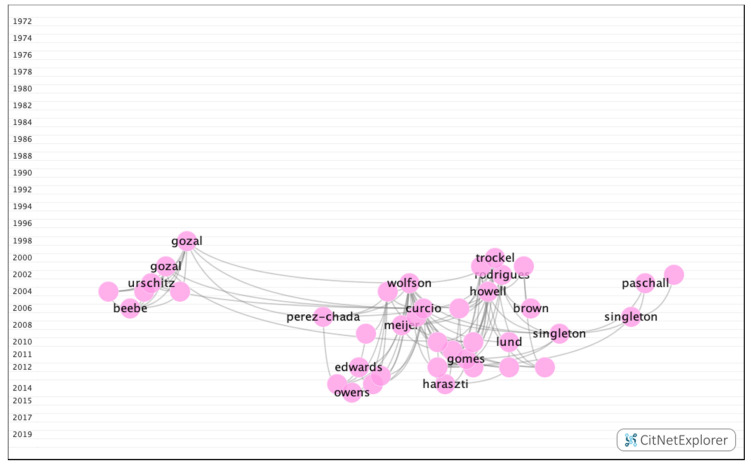
Group 7 citation network.

**Figure 14 ijerph-17-05352-f014:**
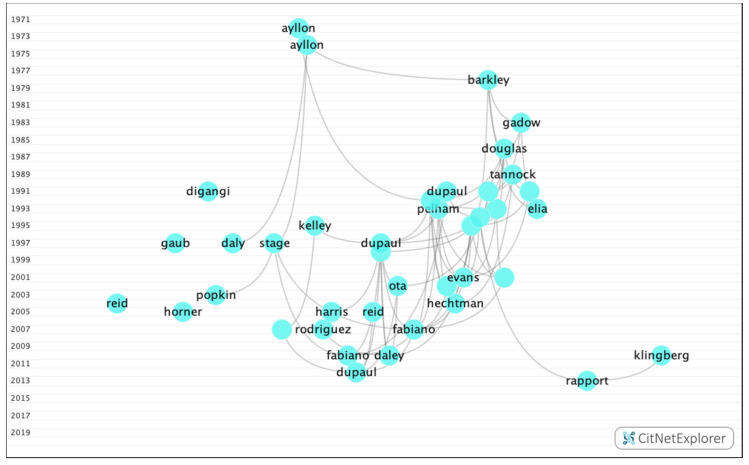
Group 8 citation network.

**Figure 15 ijerph-17-05352-f015:**
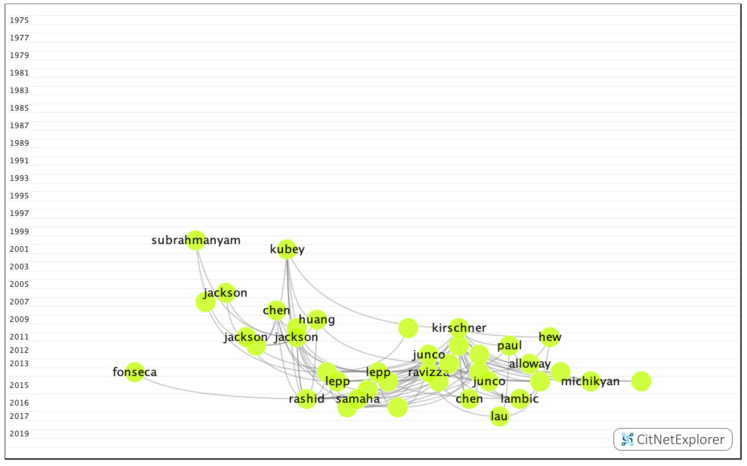
Group 9 citation network.

**Figure 16 ijerph-17-05352-f016:**
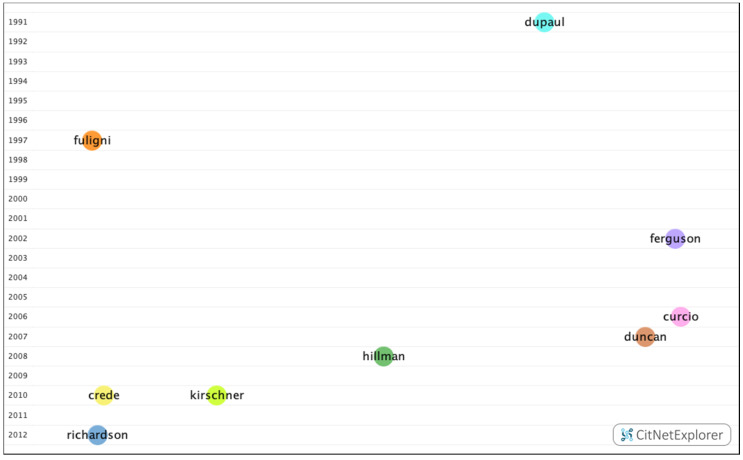
Relationship between the nine major groups.

**Figure 17 ijerph-17-05352-f017:**
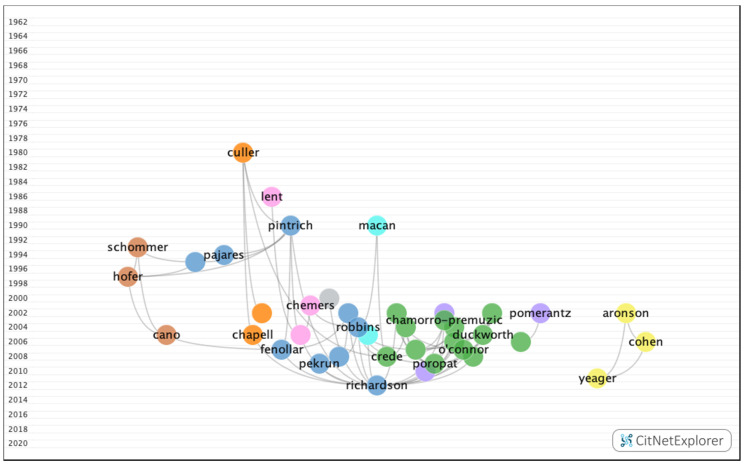
Group 1-subclusters citation network.

**Figure 18 ijerph-17-05352-f018:**
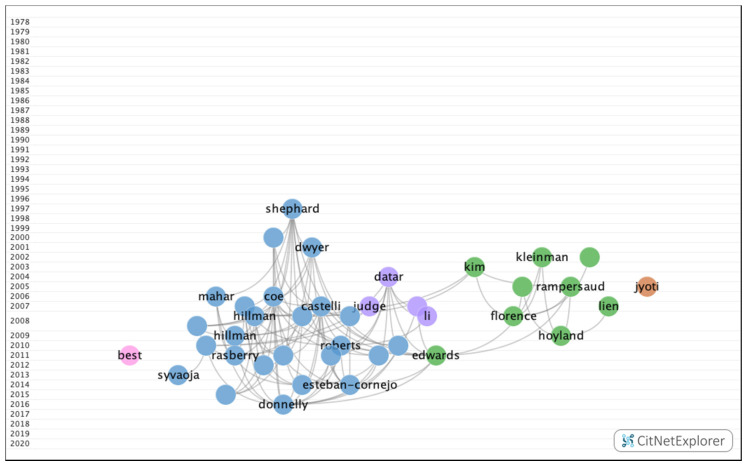
Group 2-subclusters citation network.

**Figure 19 ijerph-17-05352-f019:**
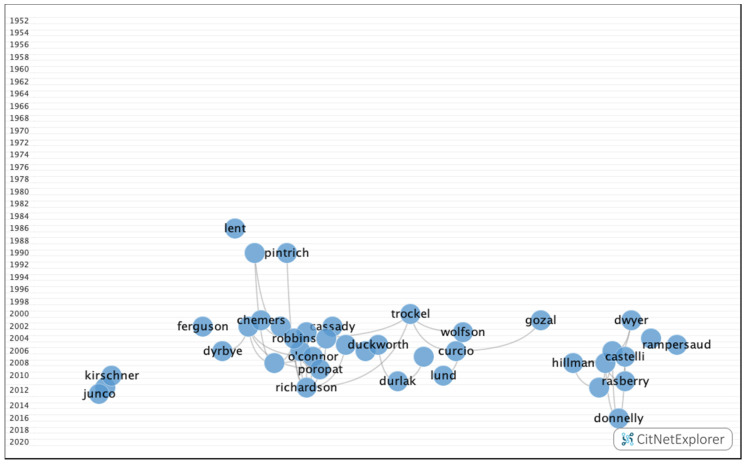
Core Function in the academic performance citation network.

**Table 1 ijerph-17-05352-t001:** Details of the 20 most cited publications on academic performance.

Author	Title	Year	Total Citation	Citation Rate
Richardson et al. [21]	Psychological correlates of university students’ academic performance: A systematic review and meta-analysis.	2012	352	50.28
Pintrich et al. [22]	Motivational and self-regulated learning components of classroom academic performance.	1990	344	11.86
Poropat [23]	A meta-analysis of the five-factor model of personality and academic performance	2009	270	24.54
Robbins et al. [24]	Do Psychosocial and Study Skill Factors Predict College Outcomes? A Meta-Analysis	2004	239	14.94
Chamorro-Premuzic et al. [25]	Personality predicts academic performance: Evidence from two longitudinal university samples	2003	179	11.19
Duckworth et al. [26]	Self-Discipline Outdoes IQ in Predicting Academic Performance of Adolescents	2005	165	11.78
O’Connor et al. [27]	Big Five personality predictors of post-secondary academic performance	2007	159	13.25
Kirschner et al. [28]	Facebook^®^ and academic performance	2010	127	14.11
Curcio et al. [29]	Sleep loss, learning capacity and academic performance	2006	126	9.69
Ferguson et al. [30]	Factors associated with success in medical school: systematic review of the literature	2002	125	7.35
Chemers et al. [31]	Academic self-efficacy and first year college student performance and adjustment.	2001	122	6.78
Kuncel et al. [32]	Academic Performance, Career Potential, Creativity, and Job Performance: Can One Construct Predict Them All?	2004	112	7.47
Hillman et al. [33]	Be smart, exercise your heart: exercise effects on brain and cognition	2008	104	9.45
Castelli et al. [34]	Physical Fitness and Academic Achievement in Third- and Fifth-Grade Students	2007	103	8.58
Trockel et al. [35]	Health-Related Variables and Academic Performance Among First-Year College Students: Implications for Sleep and Other Behaviors	2000	98	5.16
Chapell et al. [36]	Test Anxiety and Academic Performance in Undergraduate and Graduate Students	2005	95	6.78
Cassady et al. [37]	Cognitive Test Anxiety and Academic Performance	2002	94	5.53
Wolfson et al. [38]	Understanding adolescents’ sleep patterns and school performance: a critical appraisal	2003	94	5.87
Rampersaud et al. [39]	Breakfast Habits, Nutritional Status, Body Weight, and Academic Performance in Children and Adolescents	2005	94	6.71
Dyrbye et al. [40]	Systematic Review of Depression, Anxiety, and Other Indicators of Psychological Distress Among U.S. and Canadian Medical Students	2006	94	7.23

**Table 2 ijerph-17-05352-t002:** Citation network information on the nine main groups.

Main Cluster	Number of Publications	Number of Citation Links	Number of Citations Median (Range)	Number of Publications with ≥4 Citations	Number of Publications in 100 Most Cited Publication
Group 1	3223	11,097	1 (0-352)	1734	47
Group 2	1094	4407	1 (0–104)	481	22
Group 3	971	2638	1 (0–125)	494	5
Group 4	777	1356	1 (0–35)	461	1
Group 5	734	1567	1 (0–55)	456	2
Group 6	665	1107	1 (0–21)	489	2
Group 7	640	2407	1 (0–126)	256	12
Group 8	619	1587	1 (0–35)	409	0
Group 9	591	1772	1 (0–127)	329	9

**Table 3 ijerph-17-05352-t003:** The oldest and most recent journal information from the nine main groups.

Cluster		Autor	Title	Year	Total Citation
Group 1	Pioneers	Jones et al. [45]	The Individual High School as a Predictor of College Academic Performance	1962	2
	Most Recent	Carmona-Halty et al. [46]	How Psychological Capital Mediates Between Study–Related Positive Emotions and Academic Performance	2019	3
Group 2	Pioneers	Rourke et al. [47]	Neuropsychological significance of variations in patterns of academic performance: Verbal and visual-spatial abilities	1978	7
	Most Recent	Singh et al. [48]	Effects of physical activity interventions on cognitive and academic performance in children and adolescents: a novel combination of a systematic review and recommendations from an expert panel	2019	12
Group 3	Pioneers	Flook et al. [49]	Academic performance with, and without, knowledge of scores on tests of intelligence, aptitude, and personality	1968	1
	Most Recent	Hu et al. [50]	Maladaptive Perfectionism, Impostorism, and Cognitive Distortions: Threats to the Mental Health of Pre-clinical Medical Students	2019	1
Group 4	Pioneers	Finger et al. [51]	Academic performance of public and private school students	1963	1
	Most Recent	Chyn et al. [52]	Housing Voucher Take-Up and Labor Market Impacts	2018	1
Group 5	Pioneers	Henry et al. [53]	Part-time employment and academic performance of freshmen	1967	3
	Most Recent	Nordamann et al. [54]	Turn up, tune in, don’t drop out: the relationship between lecture attendance, use of lecture recordings, and achievement at different levels of study	2019	1
Group 6	Pioneers	Bewley et al. [55]	Academic-performance and social-factors related to cigarette-smoking by schoolchildren	1977	3
	Most Recent	Pörhölä et al. [56]	Bullying and social anxiety experiences in university learning situations	2019	1
Group 7	Pioneers	Lucas et al. [57]	Interaction in University Selection, Mental Health and Academic Performance	1972	3
	Most Recent	Adelantado-Renau et al. [58]	The effect of sleep quality on academic performance is mediated by Internet use time: DADOS study	2019	2
Group 8	Pioneers	Chadwick et al. [59]	Systematic reinforcement: academic performance of underachieving students	1971	2
	Most Recent	Kortekaas-rijaarsdam et al. [60]	Does methylphenidate improve academic performance? A systematic review and meta-analysis	2019	1
Group 9	Pioneers	Cooper et al. [61]	The importance of race and social class information in the formation of expectancies about academic performance	1975	1
	Most Recent	Al-Rahmi et al. [62]	Massive Open Online Courses (MOOCs): Data on higher education	2019	1

**Table 4 ijerph-17-05352-t004:** The most important subclusters from group 1.

Sub-Cluster	1	2	3	4	5
No. of publications	727	665	284	280	251
No. of citation links	1985	2447	495	704	708
Pioneers	Schultz et al., 1993 [63]	Savage et al., 1962 [64]	Greiner et al., 1997 [65]	Carrier et al., 1966 [66]	Kennelly et al., 1975 [67]
Most cited	Richardson et al., 2012 [21]	Poropat et al., 2009 [23]	Schaufeli et al., 2002 [68]	Chapell et al., 2005 [36]	Aronson et al., 2002 [69]
Most Recent	Trigeros Ramos et al., 2019 [70]	Proyer et al., 2019 [71]	Carmona et al., 2019 [46]	Alammari et al., 2018 [72]	Wang et al., 2019 [73]
Topic of discussion	Influence of motivation on academic performance	Influence of personality on academic performance	Influence of self-discipline and emotions on academic performance	Influence of anxiety on academic performance	Influence of demographic psychology on academic performance
Conclusion	Motivation has a positive influence on academic performance	Research into this group is still being carried out, therefore consensus has not yet been reached	Self-control strategies have a positive influence on academic performance	Anxiety has a negative influence on final grades; therefore self-control strategies are necessary.	Intelligence is malleable; therefore the negative stereotypes of immigrant children and academic performance must be eliminated

**Table 5 ijerph-17-05352-t005:** The most important subclusters from group 2.

Sub-Cluster	1	2	3
No. of publications	431	186	141
No. of citation links	2324	571	359
Pioneers	Nelson et al., 1993 [74]	Nidich et al., 1993 [75]	Kovacs et al., 1992 [76]
Most cited	Hillman et al., 2008 [33]	Rampersaud et al., 2005 [39]	Datar et al., 2004 [77]
Most Recent	Singh et al., 2019 [48]	Adelantado-Renau et al., 2019 [78]	Allison et al., 2019 [79]
Topic of discussion	The benefits of physical exercise on academic performance	The benefits of a healthy diet on academic performance	The link between state of health and academic performance
Conclusion	Physical exercise improves mental and physical health throughout life.	A diet that is rich in fiber, nutrients, fruits and dairy products is recommended.	Poor mental and physical health has a negative impact on academic performance

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
