# Peer review of "Current State and Future Trends: A Citation Network Analysis of the Academic Performance Field"

_ijerph, 2020, doi:10.3390/ijerph17155352_

Round 1

Reviewer 1 Report

I think your paper is of interest for readers of IJERPH.

It was easy to read, but I have some remarks:

The style of in-line references is not consistent. Sometimes Author (YYYY), sometimes only Author, and YYYY mentioned in sentence. Please unify according to guidelines of IJERPH.

In the following I'm going to  comment on specific lines (L.#):

L. 61,62 Please specify what you mean by “connection between publications and authors”.

L. 65 you probably mean more precisely: “limiting the search to a topical search (TS) in article, keywords and abstract”

L. 66, 81 The first publication found by the topical search is from 1930(!) and there are six papers before 1952. Why do you only start in 1952? Please explain.

L. 71 what are the other databases and what are you doing with them?

L.77-79: Do you mean, that you as authors tuned the parameter “minimum number of citation links”?

L. 83 with “on the web” you probably mean the “inside the WoS”?

L. 99 table 1: apart from the journal title announced in the header, no further specification is needed here, because all papers are also listed in the References.

But it would be better to number the references, so that in

L. 100 ff. the subsets of papers could be specified.

L. 119 please give the clustering parameters as resolution and minimal cluster size.

L. 198 please explicate how you analyzed the relationship between the groups.

L. 234,235 the second reference [81] is from 1990 and therefore no “first piece”!

L. 238 How did you identify 2018 as key year? Having the highest number of publications is not enough. Or maybe you mean 2016, because the ref. [82,83] are from 2016?

l. 274 you need to insert : … as this could <result> in ….

L. 282 conclusions unusually short; how did you prove that “all” of relevant publications were covered?

Reviewer 2 Report

The paper Current State and Future Trends: A Citation Network Analysis of the Academic Performance Field brings an original citation analysis of “academic performance” field, despite it doesn’t present an inovative methodology to justify its publication.

The results are presented in a descriptive way. In this sense, I would like to suggest a deeper discussion regarding countries and journals. Why that countries are the most productive in this field? Why that journals are used to publish the papers? The citation results could also be more discussed, maybe in a historical analysis over authors and topics, the field development over the years.

At the methological procedures, the search on Web of Science must be detailed with information regarding wich index was used (Science Citation Index or the others), the kind of search used (if simple interface or complete interface), download date. It will be great if authors could explain why they didn’t use other expressions/words on the search, like “student performance”, “cognitive engagement” , “academic efficacy” and so on. If “academic performance” is the expression most common it should be stated.

Citation studies generaly make a clean up process on authors names. Have this paper did any correction on the names?

The paper doesn’t present a conclusion.

Reviewer 3 Report

Brief Summary

The paper employs a citation network analysis to examine the makeup of literature on “academic performance” from 1952 – 2019, drawing bibliographic records from the Web of Science Database. Specifically, the study looks at links between publications and authors via the aforementioned citation networks. The study is somewhat exploratory, aiming to identify different research areas within the academic performance literature and identify the most cited publications rather than raising and answering more specific research questions. Furthermore, the study demonstrates the type of overview of a field one can acquire through citation network analysis, as opposed to more conventional approaches (e.g. traditional literature review).

Broad Comments and Specific Comments

The main contribution of this paper is in illustrating how citation network analysis can be used to provide an overview of a particular area of research, in this case, academic performance research. This differs from a more traditional literature review in that, by using this methodology, researchers can come closer to examining a field comprehensively rather than selectively. Of course, the tradeoff here is that the level of detail that can be provided about the literature is vastly diminished. This doesn’t weigh for or against one approach over the other, but is simply a feature of different tools being appropriate for different tasks.

Of particular interest is the identification of 23 groups within the publications studied, 9 of which contained a significant number. Unlike some of the information provided in the paper, this grouping is a discovery that is unlikely to be discovered using more conventional research methods. In fact, this seems to be the main discovery offered by the paper, as the other information provided could have been gleaned from analyzing the Web of Science data in a more traditional format (e.g. publications per year, top ten journals with most publications).

While the approach taken by the authors is interesting, there are a few major issues with the paper. First, the Materials and Methods section is short on detail. Given that this is, to some extent, a methodological paper, I would expect more elaboration on the details of the methodology employed, and justification for this approach over more conventional methods. This sentiment isn’t due to skepticism about the approach taken, but rather from the belief that the paper would make a more significant contribution by offering such elaboration.

The authors report that they used the Citation Network Explorer software to analyze and visualize the literature under consideration, but not much is reported regarding what this analysis consists in. For instance, beginning at line 69 the authors write, “By using Citation score as an attribute, a quantitative analysis of the most cited publications within a specific time interval was performed. Through this, internal connections with the WOS database were quantified. By considering other databases, the external connections were also quantified.” Here it would be helpful to explain what is meant by “internal” and “external” connections. I take it this means citations, but are these mutual citations, common citations, etc.? The authors also state that the clustering function was used to establish groups for each publication and that central publications were analyzed using the Identifying Core Publications function. It would be beneficial to provide some detail as to how these functions operate.  

The discussion of the nine main groups discovered in the literature is interesting, though it is quite brief and surface level in all cases. It would be interesting to see how the authors think this data could be used to further research in the relevant subject areas. What sorts of research questions does this data enable? How insights might it provide that would remain undiscovered by the use of other methods? How might using citation network analysis change research within the field(s)?

In summary, this paper and the study it reports strike me as incomplete. The authors provide an interesting set of preliminary data, but it is presented with inadequate discussion or argument for its significance. I encourage the authors to consider more in the way of what specific points they want to convey in this study. If the aim is to indeed simply provide some preliminary data, then I’m not sure traditional journal publication is the best route of disseminating this information. It would be more beneficial to provide a fleshed-out dataset with more detailed visualizations through publication in a data repository or data journal. But as it stands, I do not recommend this paper for publication.

Round 2

Reviewer 3 Report

The revision does a particularly nice job of expanding on the methods employed, providing more details about both the way in which Web of Science and the Citation Network Explorer were used. The authors have also provided a bit of additional discussion of the various groups they identified via the citation network analysis, though I still find it to be pretty brief and surface level, not addressing some of the questions raised in my first report. 

The paper still lacks a discussion that connects the subject matter (academic performance) to the methodology employed. Most of the introduction consists in a lit review of the academic performance literature with one paragraph about citation network analysis. The authors need to explain why the later is a good method to explore the former. There is some mention of this in the data analysis section, but it should be mentioned in the introduction. Moreover, the discussion in the data analysis section doesn't touch on why citation analysis is a good fit for examining academic performance research. 

The discussion section should also dedicate more space explaining how the insights reported were enabled by the citation analysis approach. Similar concerns arise for the conclusion. 
